# 3D covalent organic framework membrane with fast and selective ion transport

Tianhao Zhu[1,2,4], Yan Kong[1,2,4], Bohui Lyu[3], Li Cao [1,2], Benbing Shi[1,2], Xiaoyao Wang [1,2], Xiao Pang[1,2], Chunyang Fan[1,2], Chao Yang [1,2], Hong Wu [1,2] & Zhongyi Jiang [1,2,3] ✉

3D ionic covalent organic framework (COF) membranes, which are envisioned to be able to break the trade-off between ion conductivity and ion selectivity, are waiting for exploitation. Herein, we report the fabrication of a 3D sulfonic acid-functionalized COF membrane (3D SCOF) for efficient and selective ion transport, using dual acid-mediated interfacial polymerization strategy. The 3D SCOF membranes possess highly interconnected ion transport channels, ultramicroporous pore sizes (0.97 nm), and abundant sulfonate groups (with a high ion exchange capacity of 4.1 mmol g$^{-1}$), leading to high proton conductivity of 843 mS cm$^{-1}$ at 90 °C. When utilized in osmotic energy conversion, a high power density of 21.2 W m$^{-2}$, and a remarkable selectivity of 0.976 and thus an exceptional energy conversion efficiency of 45.3% are simultaneously achieved. This work provides an alternative approach to 3D ionic COF membranes and promotes the applications of 3D COFs in ion transport and separation.

Ion-conductive membranes which allow selective and efficient ion transport are highly desirable for electrochemical energy-related applications, such as fuel cells, electrodialysis, and osmotic energy conversion[1–4]. The ion conductivity and selectivity jointly determine the electrochemical system performance in terms of power density and energy efficiency[5–7]. However, there often exists a trade-off between ion conductivity and selectivity of the membranes, as the selectivity relies on smaller pore size than that of hydrated ions which in turn smaller pore size leads to a remarkable decrease in ion conductivity. In recent years, a variety of polymer membranes featuring dense structures and small aperture channels have been developed for selective ion transport, which show a high ion selectivity arising from the size sieving and electrostatic interaction[8–10]. However, the long and discontinuous channels formed by the self-assembly of flexible chain segments hinder ion transport to a large extent[11]. The past few years have witnessed intensive efforts in emerging 2D materials as ion-conductive materials, such as graphene oxide, transition metal carbides and nitrides, and molybdenum disulfide[12–15]. The precise control

of interlayer spacing is expected to achieve high ion selectivity[16,17]. However, the tortuous pathways through the space between nanosheets and insufficient charge density often lead to high mass transfer resistance and insufficient ion conductivity. Thus, fabrication of ionic membranes with simultaneous high conductivity and selectivity remains a great challenge[18–22]. In this respect, the porous membrane with tunable pore structure and designable channel surface is a promising platform to produce advanced membranes[23].

As an intriguing type of porous crystalline material, ionic covalent organic framework (COF) membranes have displayed huge potential as high-performance ion conductors due to the permanently ordered channels and precise arrangement of ionic groups[24,25]. Based on the covalent connectivity in different dimensions, COFs are divided into two-dimensional (2D) structures and three-dimensional (3D) networks. Over the past decade, studies in state-of-the-art COF membranes mainly focus on 2D COFs[26–28]. However, 2D COFs usually have relatively large pore sizes of 1 − 5 nm, making it difficult to realize highly selective ion sieving, despite that abundant in-plane pores within the

[1]Key Laboratory for Green Chemical Technology of Ministry of Education, School of Chemical Engineering and Technology, Tianjin University, Tianjin 300072, China. [2]Haihe Laboratory of Sustainable Chemical Transformations, Tianjin 300072, China. [3]Joint School of National University of Singapore and Tianjin University, International Campus of Tianjin University, Binhai New City, Fuzhou 350207, China. [4]These authors contributed equally: Tianhao Zhu, Yan Kong. ✉e-mail: zhyjiang@tju.edu.cn

nanosheets are favorable for high ion conductivity. By contrast, 3D COFs have narrow pore sizes in the range of ~0.5–1.5 nm due to the interpenetration of 3D frameworks, offering a promising avenue to separate ions[29]. Moreover, 3D COFs have 3D open spatial structures and multidirectional, interconnected networks, resulting in more efficient ion transport compared with 2D COFs[30–33]. Consequently, 3D COFs are recognized as promising candidates for ion conductors and separators to break the trade-off between ion conductivity and ion selectivity[34,35]. However, due to the challenges in the assembly of nonplanar building blocks and the control of film-forming processes, only a few 3D COF membranes have been reported[36,37]. Moreover, the tetrahedral and ionic groups-functionalized building blocks tend to have larger steric hindrance, lower diffusion rate and lower reactivity, which bring about additional difficulty to fabricate 3D ionic COF membranes. Especially, the preparation of 3D ionic COF membranes, featuring interconnected micropores and abundant ion groups, has not been reported.

Herein, we demonstrated the fabrication of 3D ionic COF membranes for selective and efficient ion transport, using a dual acid-mediated interfacial polymerization strategy. 3D sulfonic acid-functionalized COF (3D SCOF) membranes were prepared via reactive assembly of sulfonic acid-functionalized aldehyde monomers and tetra-(4-anilyl) methane monomers in a dual acid-containing organic-water two-phase system. The 3D SCOF membranes possessed fully extended and highly interconnected transport channels, sub-1-nm ultramicroporous pore sizes, and abundant sulfonate groups, which afforded rapid transport of protons and cations as well as high selectivity towards anions. The 3D SCOF membranes exhibited high proton conductivity of 843 mS cm$^{-1}$ at 90 °C, 100% relative humidity (RH). As a proof-of-concept application, the ion permselectivity of 3D SCOF membrane is evaluated by osmotic energy conversion. By mixing artificial sea water and river water, the 3D SCOF membranes exhibited a high power density of 21.2 W m$^{-2}$, more than 4-fold larger than the commercialization benchmark, and a record-high ion selectivity of 0.976 and thus an exceptional energy conversion efficiency of 45.3% simultaneously. Furthermore, the power density reached 69.6 W m$^{-2}$ under a 500-fold salinity concentration gradient, indicating the membrane applicability in hypersaline environments.

## Results

As shown in Fig. 1a, a de novo design of 3D ionic COF was demonstrated. 3,3'-((2,5-diformyl-1,4-phenylene) bis(oxy)) bis (propane-1-sulfonic acid) monomer (monomer A) was synthesized from 2,5-dihydroxyterephthalaldehyde and assembled with the tetrahedral tetra-(4-anilyl) methane monomer (monomer B) to generate the side-chain sulfonic acid-functionalized 3D frameworks (3D SCOF). A dual acid-mediated interfacial polymerization strategy was invented to prepare 3D SCOF membrane, in which the monomer A was dissolved in acetic acid aqueous solution while the monomer B was dissolved in octanoic acid (Fig. 1c and Supplementary Fig. 1). The organic phase containing amine monomers was added slowly onto the top of the aqueous phase containing aldehyde monomers to form a bilayer and the interfacial reaction was kept at 20 °C for 3 days in undisturbed condition. Then we collected the self-standing 3D SCOF membrane formed at the interface and washed it thoroughly with tetrahydrofuran, acetone, water and ethanol in turn. The diameter of the membrane was estimated about 3.5 cm (Fig. 1d), which was determined by the size of the reaction vessel used. The as-prepared 3D SCOF membrane was intact, defect-free, flexible and mechanically robust (Fig. 1e, f, Supplementary Fig. 2). As shown in Supplementary Fig. 3, the 3D SCOF membrane possessed a high Young's modulus of approximately 2.4 GPa on average. Moreover, the membrane can be preserved in both wet and dry state (Supplementary Fig. 4). The scanning electron microscopy (SEM) images revealed the tight structure and uniform thickness

throughout the membrane (around 1 μm). The transmission electron microscope (TEM) EDS-mapping images demonstrated that the carbon, oxygen, nitrogen and sulfur elements were uniformly distributed in the 3D SCOF membrane, which confirmed the homogeneous structure of the membrane (Fig. 1g).

Fourier transform infrared (FTIR) spectra of the as-synthesized 3D SCOF membrane clearly revealed the formation of characteristic bands of the framework structure (Fig. 2a). Compared with the aldehyde and amine monomers, 3D SCOF showed typical C=N imine stretching bands at 1623 cm$^{-1}$ and disappearance of H−C=O and N−H stretching bands at 1708 cm$^{-1}$ and 3194 cm$^{-1}$, which confirmed the complete consumption of monomers and formation of imine linkages[38,39]. In addition, the FTIR spectra of 3D SCOF verified the presence of O=S=O peaks of sulfonic acid groups at 1078 cm$^{-1}$, manifesting the abundant negatively charged sites within the frameworks. In accordance with the FTIR spectra, the solid-state $^{13}$C NMR measurement also confirmed the characteristic peaks of imine bonds at 157 ppm (Fig. 2b). Moreover, a sharp and distinct peak at 51 ppm arising from the carbon atoms that the sulfonic acid groups attached could be observed in the NMR spectra, further confirming the existence of sulfonic acid groups in the 3D SCOF membrane[40,41]. We further determined the ion exchange capacity (IEC) (with a value of 4.1 mmol g$^{-1}$) of 3D SCOF membrane by titration, verifying the presence of high-concentration sulfonic acid groups. Thermal gravimetric analysis (TGA) (Supplementary Fig. 5) revealed the 3D SCOF was stable up to 350 °C under nitrogen atmosphere, suggesting the high thermal stability.

The crystal structure of the 3D SCOF membrane was resolved by the powder X-ray diffraction (PXRD) measurement in conjunction with structural simulations (Fig. 2c). The experimental PXRD pattern with intense diffraction peaks demonstrated the ordered structure of the 3D SCOF membrane. After a geometrical energy minimization by using the universal force field based on the 3-fold interpenetrated dia frameworks, the unit cell parameters of 3D SCOF were obtained ($a = b = 34.1$ Å, $c = 13.1$ Å, and $\alpha = \beta = \gamma = 90°$) (Supplementary Table 1). The simulated PXRD pattern of 3D SCOF matched well with the experimental result. Peaks at 5.2°, 9.0°, 10.4° and 14.5° for 3D SCOF corresponded to the (200), (211), (400) and (112) Bragg peaks of space group I41/A[42–44]. The unit cell parameters were nearly equivalent to the simulated one with good agreement factors of Rwp = 10.49% and Rp = 8.09%. Some alternative structures, such as 1- and 2-fold interpenetrated dia nets, were also set up; however, their simulated PXRD patterns did not match the experimental result (Supplementary Fig. 6). According to these results, the 3D SCOF was proposed to adopt a 3-fold interpenetrated dia framework (Fig. 1b and Supplementary Tables 1). Notably, the 3D SCOF exhibited less interpenetrated porous frameworks than most of the 3D COF materials, which could be ascribed to the steric hindrance and electrostatic repulsion derived from the long sulfonic acid-functionalized side chains[45–47]. In addition, the high crystallinity of the 3D SCOF membrane was also verified by the lattice patterns all over the visual field in the high-resolution TEM image (Fig. 1h). Particularly, the zonary bright regions revealed the highly ordered pores of the 3D framework structure in the membrane. Nitrogen adsorption-desorption isotherm was conducted at 77 K to evaluate the porosity of the 3D SCOF membrane (Supplementary Fig. 7). The Brunauer-Emmett-Teller (BET) surface area for 3D SCOF membrane was calculated to be 49 m$^2$ g$^{-1}$. The low surface area for the membrane may be due to the following two factors: (1) the long functional side chains hampered the accessibility of nitrogen molecules; (2) the tight structure of the membrane acted as an additional barrier for nitrogen molecules[47–49]. The pore size of 3D SCOF membrane was predicted to be 0.97 nm based on the diameter of the largest included sphere in the framework. To confirm the channel size of the 3D SCOF membrane, the rejection tests of polyethylene glycols (PEG) with various molecular weights were conducted on a dead-end filtration device. As shown in Supplementary Fig. 8, the mean channel

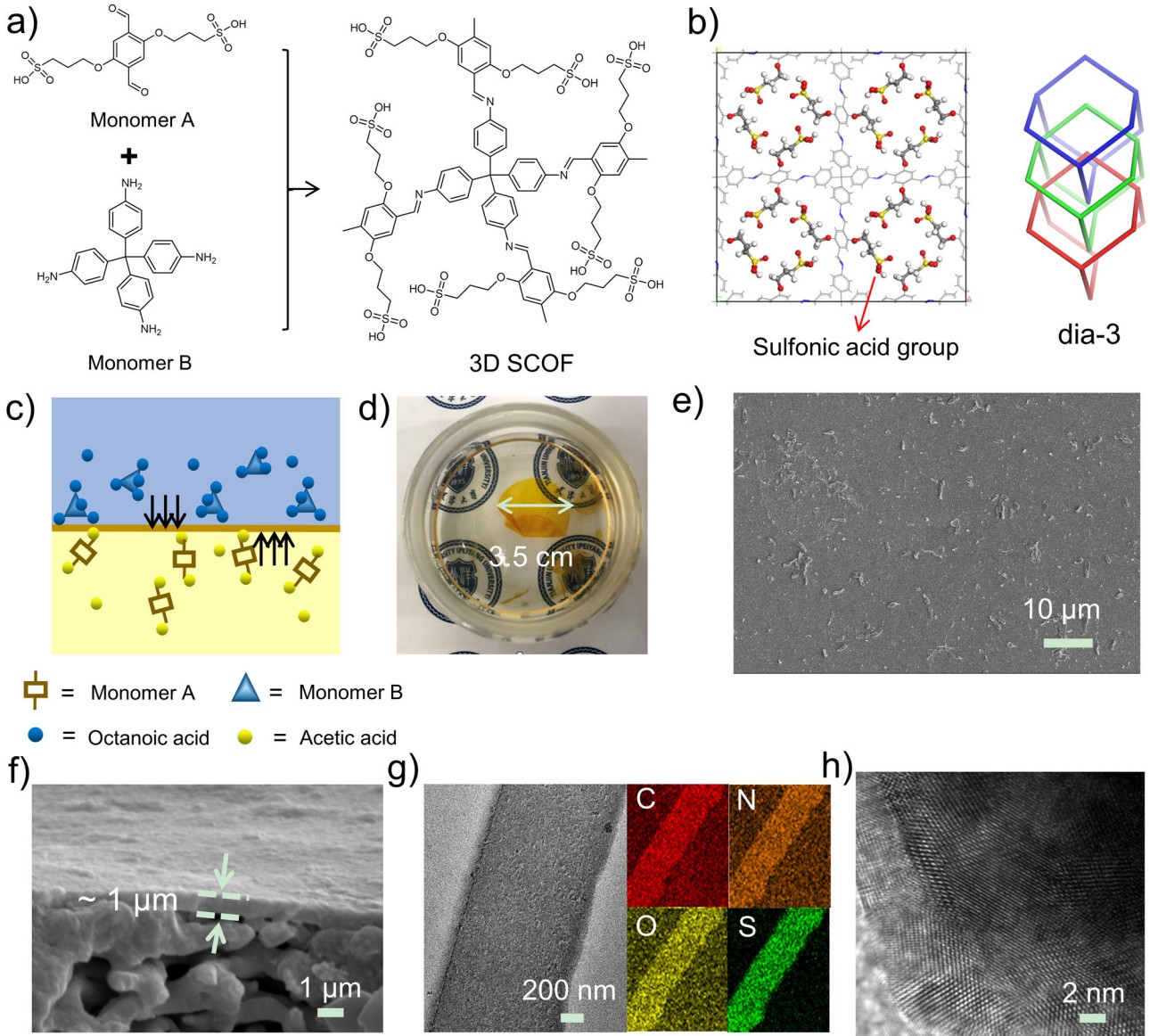

**Fig. 1 | Preparation and morphological characterization of 3D SCOF membrane.**
**a** Schematic representation of the synthesis of 3D SCOF via imine condensation.
**b** The structural representation of 3D SCOF. Left: the structure viewed parallel to the channels; right: a representative model of 3-fold interpenetration. **c** Schematic of the dual acid-mediated interfacial polymerization process for preparing 3D SCOF membrane. **d** Photograph of a 3D SCOF membrane (soaked in water). **e** Top-view SEM image of the 3D SCOF membrane. **f** Cross-section SEM images of the 3D SCOF membrane. **g** TEM EDS-mapping images of the 3D SCOF membrane. **h** The high-resolution TEM image of the 3D SCOF membrane.

diameter of the 3D SCOF membrane calculated based on PEG rejection was about 0.98 nm, which was consistent with the predicted value.

In the dual acid-mediated organic-aqueous reaction system, amine monomers were dissolved in the top organic phase consisting of one acid and then reacted with the acid to form amine-acid complex because of the formation of hydrogen bonds (quantitively described by the interaction energy). Similarly, aldehyde monomers were dissolved in the bottom aqueous phase containing one water-soluble acid and then combined with the acid to form aldehyde-acid complex[46]. Subsequently, the amine-acid complex and aldehyde-acid complex diffused in the opposite direction and began to react where they meet. After the breakage of hydrogen bonds in these two kinds of complexes, the free amine and aldehyde groups began to react and imine bond formed[46,50,51]. As the reaction progressed, lots of microcrystals formed and connected to assemble into the membranes. To elucidate the effect of acid on the growth process of 3D COF membrane, we executed different dual-acid interfacial polymerization processes by

changing the types of acids in the aqueous phase and organic phase respectively.

Firstly, different acids such as trifluoroacetic acid (TFA), p-toluene sulfonic acid (PTSA), and acetic acid (AA), were employed in the aqueous phase to explore their effects on the membrane formation process. Octanoic acid (OA) was fixed as the organic solvent (Supplementary Fig. 9). For the TFA/OA system, after reaction for 7 days, no films formed at the interface and a mass of aggregates of particles were collected in the organic phase, indicating that the reaction took place predominantly in the organic phase (Supplementary Fig. 10a and b). For the PTSA/OA system, the reaction was slowed down significantly and only some film fragments with many defects formed on the aqueous phase near the interface (Supplementary Fig. 10c and d). By contrast, for the AA/OA system, a continuous, flexible and defect-free membrane (3D SCOFM-AA/OA) was obtained at the interface. With the introduction of acids in the aqueous phase, the reactivity of monomer A increased, that is, monomer A became

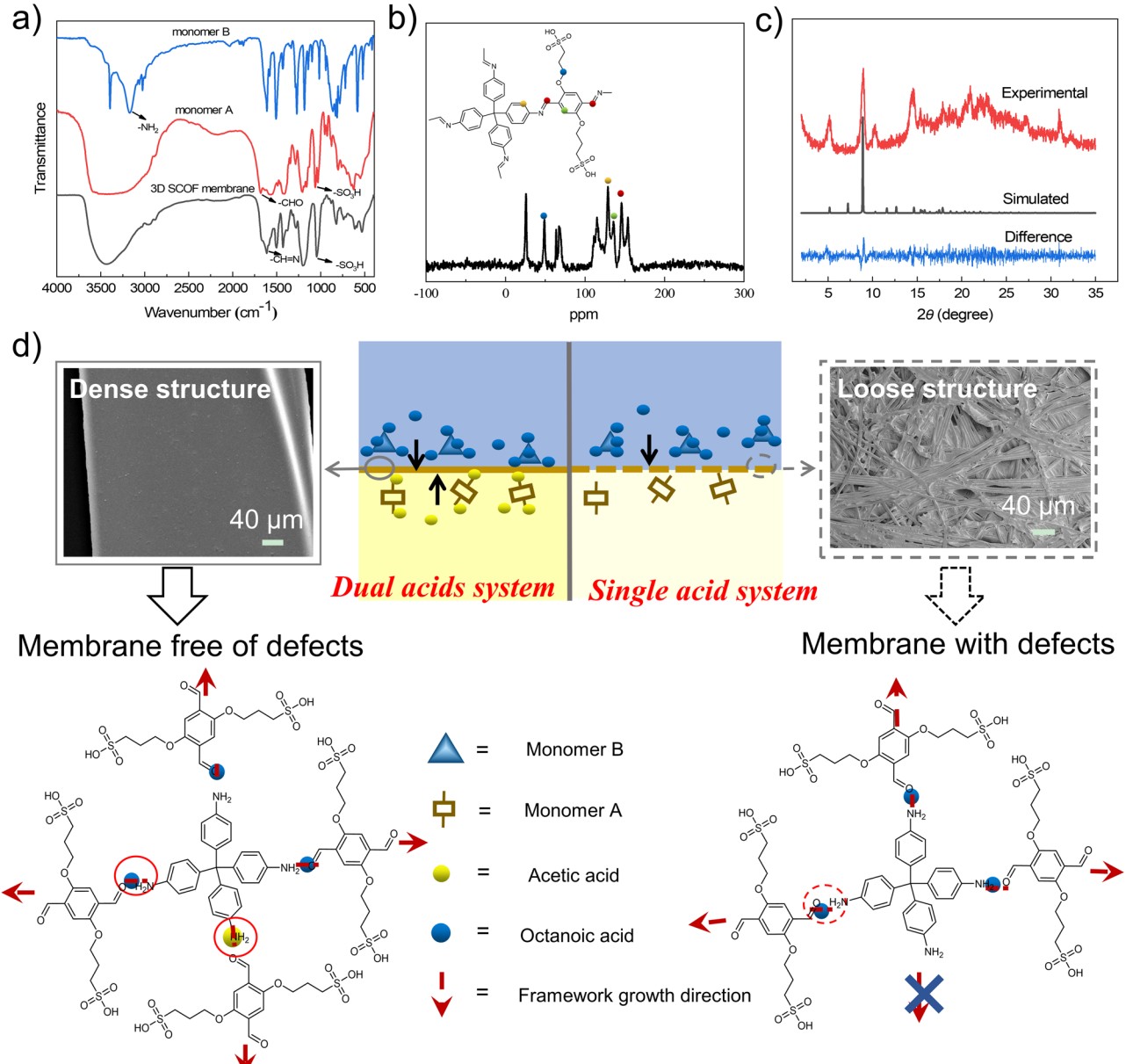

**Fig. 2 | Structural characterization of 3D SCOF membrane and the schematic for membrane growth mechanism. a** FTIR spectra of the monomer A, monomer B and corresponding 3D SCOF membrane. **b** $^{13}$C CP-MAS solid-state NMR spectra of the 3D SCOF membrane. **c** PXRD patterns of 3D SCOF membrane. The experimentally observed patterns are shown in red, the calculated pattern in black, and their difference in blue. **d** Schematic for the membrane growth mechanism of single-acid and dual-acid interfacial polymerization process.

more prone to undergo nucleophilic addition reaction because of the protonation of aldehyde group (quantitively described by Fukui function, Supplementary Figs. 11–14). However, the increase in monomer reactivity did not necessarily lead to an improvement in the interfacial morphology of membrane. It was speculated that the addition of acids also had a great influence on the diffusion process of monomers in the organic-water two-phase system. We simulated the diffusion behaviors of three kinds of acids in the biphasic system based on the molecular dynamics method and calculated the interaction energies of these acids with monomer A via the first-principles density functional theory approach (Supplementary Table 2). As shown in Supplementary Movie 1–3, it can be clearly observed that TFA was easy to diffuse from water phase to organic phase, AA had a moderate diffusion rate to organic phase, while PTSA was difficult to diffuse to organic phase across the interface. Due to the interactions of monomer A with corresponding acid (or the formation of aldehyde-acid

complex), the diffusion behavior of acid also affected that of monomer A. Especially for the TFA/OA system, since TFA diffused rapidly from aqueous phase to organic phase, which also brought the diffusion of a large number of aldehyde monomers to organic phase, the reaction zone shifted from the interface region to the organic phase. By contrast, for the PTSA/OA system, due to the slow diffusion rate of PTSA and aldehyde monomers to be affected through the interface, the reaction was slowed down to a great extent and took place mainly in the aqueous phase, making it quite difficult to obtain an intact film. For the AA/OA system, due to the appropriate diffusion rate of acetic acid in the reaction system, by coordinating the diffusion and reaction process of monomers, the reaction was confined in the interface region and thus a continuous and tight membrane was obtained at the interface (Supplementary Fig. 15).

Subsequently, based on the above experimental results, we chose acetic acid solution as aqueous phase, different acids such as hexanoic

acid (HA), octanoic acid (OA), and nonanoic acid (NA) were employed as the organic phase to further analyze their effects on the membrane growth process. For the AA/HA and AA/DA system, we could only collect some film fragments with many defects at the interface (Supplementary Fig. 16). By contrasting the interaction energies of several complexes, it was observed that the uniform and defect-free membranes could be obtained when the interaction energy of amine-acid complex was close to that of aldehyde-acid complex (Supplementary Fig. 17). It was deduced that the difficulty of hydrogen bond breakage in these two complexes would affect the extent of reaction of aldehyde and amino groups of monomers, and then affect the three-dimensional growth of the framework. These results illustrated that the interaction energies of monomer-acid complexes had a pivotal role in regulating the diffusivity of monomers and close interaction energies of aqueous complex and organic complex was essential to preparing intact and highly crystalline 3D COF membranes.

We also conducted the single-acid interfacial polymerization processes: octanoic acid as the organic phase and water as the aqueous phase, or mesitylene as the organic phase and acetic acid solution as the aqueous phase. In the single-acid system, thin films were formed at the interface and were difficult to transfer onto other substrates without damaging their structure. Specifically, for the single-octanoic acid system, the resultant film was assembled by the disordered fiber-like crystallites along with many defects and voids (Fig. 2d). For the single-acetic acid system, the resultant film was formed by the accumulation of flower-like spherical nanoparticles and displayed a rough and defective surface (Supplementary Fig. 18). This further illustrated that the dual acid-mediation played an important role in the structural morphology of the resultant COF membranes.

To evaluate the ion transport performance of the 3D SCOF membrane (3D SCOFM-AA/OA), the proton conductivity was acquired by using electrochemical impedance spectroscopy based on an electrochemical workstation (Supplementary Fig. 19)[14,52]. The proton conductivity of the 3D SCOF membrane increased with the temperature and achieved a maximum proton conductivity of 843 mS cm$^{-1}$ at 90 °C, 100% RH (Fig. 3a and Supplementary Fig. 20). To explore the proton conduction mechanism, we calculated the activation energy of the 3D SCOF membrane based on the temperature-dependent conductivity profiles (Supplementary Fig. 21). The activation energy of the membrane was 0.16 eV, which indicated that the Grotthuss mechanism played a dominant role in proton conduction[53]. Due to the extended, interconnected channels and the high concentration of shuttling anion sites (sulfonic acid groups), the 3D SCOF could construct a highly continuous hydrogen-bonded network for efficient proton conduction under the high-humidity condition. We further made a comparison between proton conductivity of the 3D SCOF membrane with the state-of-art COFs or MOFs-based proton conductors and proton exchange membranes (PEMs) reported in literature (Fig. 3b). The proton conductivity of the 3D SCOF membrane reached up to 3 times the proton conductivity of the state-of-the-art PEMs, highlighting the great potential of our 3D SCOF membrane for high-efficiency ion transport. In addition, we evaluated the proton conduction stability test of the 3D SCOF membrane at 60 °C, 100% RH for a week (Supplementary Fig. 22). The proton conductivity of the membrane remained almost unchanged, which demonstrated the stable structure of the membrane.

We further investigated the osmotic energy conversion capability of the 3D SCOF membrane (Fig. 3c). To increase the operability, we transferred the 3D SCOF membrane to a polyacrylonitrile substrate. The transmembrane ionic transport properties of the 3D SCOF membrane were examined by current−voltage ($I - V$) measurements in a two-compartment electrochemical cell (Supplementary Fig. 23)[54]. Figure 3d showed the representative $I - V$ curve of the 3D SCOF membrane under the standard artificial river water (0.01 M NaCl) and sea water (0.5 M NaCl) system. The net ionic current of 4.8 μA at $V = 0$

indicated a net cation flux from the high-concentration side to the low-concentration side and reflected a cation selectivity of the 3D SCOF membrane. A salt bridge was used to eliminate the redox potential of the electrodes, so we could directly obtain the diffusion potential ($E_{diff}$) from the intercept on the voltage axe of the $I - V$ curves. The $E_{diff}$ value of the 3D SCOF membrane reached 88.9 mV, which was much higher than those of previously reported membranes. Furthermore, the corresponding cation selectivity (quantitatively described by cation transference number, $t$) was calculated to be 0.976, which was close to the ideal cation selectivity ($t = 1$). The high cation selectivity was attributed to the efficient overlap of the electric double layers caused by the highly negatively charged ultramicroporous channels of the 3D SCOF membrane (0.97 nm), as estimated from the length of the Debye screening layer. The double Debye length in 0.5 M and 0.01 M NaCl were 0.86 and 6.1 nm. Therefore, the corresponding energy conversion efficiency achieved up to 45.3%, which is the highest value reported.

To evaluate the application prospects of the 3D SCOF membrane, the harvested osmotic energy was used to supply an external resistance load ($R_L$) under a 0.01 M/0.5 M NaCl concentration gradient (Fig. 3e and Supplementary Fig. 24). It should be noted that the effective testing area of 3D SCOF membrane was 0.03 mm$^2$, the same as previous reports. The current densities ($I_{osmosis}$) were recorded by changing $R_L$, and the output power densities ($PD$) were accordingly calculated. The power density reached a maximum value of 21.2 W m$^{-2}$ at a load resistance of about 7 kΩ (comparable to the internal resistance of the device), which exceeded most reported 1D and 2D nanofluidic channel membrane systems to date and was over four times the value of the commercialization benchmark of ~5 W m$^{-2}$. The high power density benefited from the highly interconnected pores and thus additional and shorter ion transport pathways. In contrast, the output power density of Nafion membrane, the representative commercial cation exchange membrane, was estimated. Nafion membrane exhibited a peak power density of 6.4 W m$^{-2}$, much lower than that of 3D SCOF membrane, confirming the crucial role of abundant interconnected channels. Furthermore, we evaluated the long-term stability of the membrane in osmotic energy conversion at a 50-fold concentration gradient (Supplementary Fig. 25). A negligible decrease was observed in the measured power density after 6 days, which can be ascribed to the good chemical and physical structural stability of the 3D SCOF membrane. The observation clearly demonstrated the great application reliability of the 3D SCOF membrane. We also tested the output power density of the 3D SCOF membrane-based osmotic power generator under the 5-fold and 500-fold concentration gradients. The maximum power density can achieve ~69.6 W m$^{-2}$ at the 500-fold concentration gradient, indicating the application viability of the 3D SCOF membrane in hypersaline environments (Supplementary Fig. 26). In addition, the osmotic energy production using other typical types of chloride salts were also explored. It was found that the power densities follow the trend of KCl (28.6 W m$^{-2}$) > NaCl (21.2 W m$^{-2}$) > LiCl (15.5 W m$^{-2}$) > CaCl$_2$ (13.8 W m$^{-2}$), which was consistent with the order of diffusion coefficients of cations (Supplementary Fig. 27). As the 3D SCOFM was cation-selective, the faster the cation diffused, the larger the ionic current was. The comparison with previously reported membranes was shown in Fig. 3f from output power density and energy conversion efficiency. Our 3D SCOF membrane showed a remarkable energy conversion efficiency of 45.3%, which was the highest among all existing materials utilized in osmotic energy conversion, with a simultaneously high output power density of 21.2 W m$^{-2}$, outperforming most of state-of-the-art membranes. It should be noted for 2D COF membrane, the abundant ordered channels of 2D COF membrane were conducive to ion diffusion for achieving high power density, but the relatively larger pore sizes than 1 nm generated inadequate ion selectivity and thus

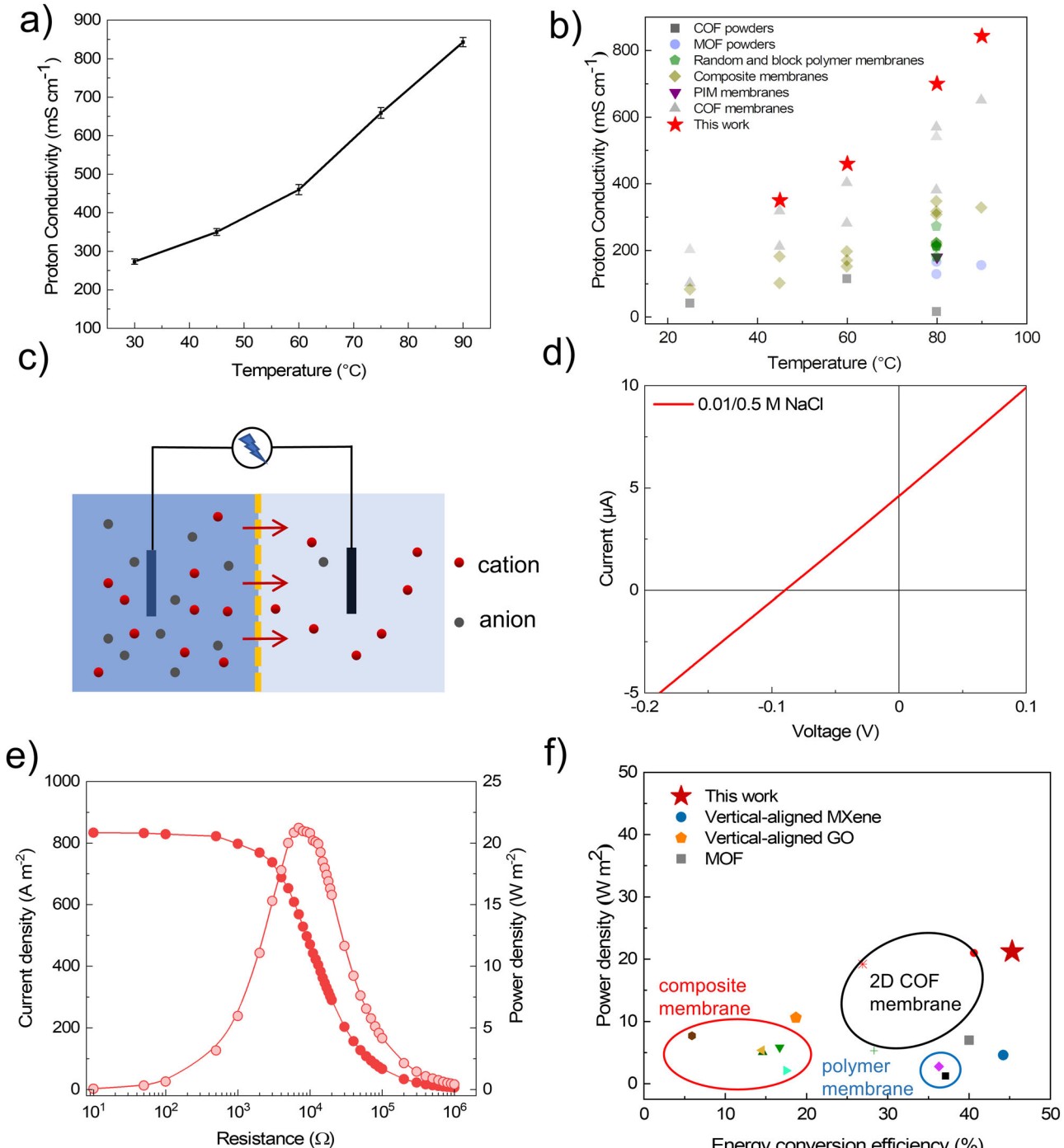

**Fig. 3 | Proton conduction and osmotic energy conversion performance of 3D SCOF membrane. a** Temperature-dependent proton conductivity of the 3D SCOF membrane under 100% RH. The error bars in the figure represent standard deviations for three measurements. **b** Comparison in proton conductivity of the COFs or MOFs-based proton conductors, polymer-based PEMs in literature and 3D SCOF membrane in our work. Corresponding ion-conducting materials were shown in Supplementary Table 3. **c** Schematic of the osmotic energy conversion process.

**d** $I−V$ curve of the 3D SCOF membrane under a 0.01 M/0.5 M NaCl concentration gradient. **e** Current density and power density of 3D SCOF membrane as functions of external resistances under a 50-fold concentration gradient. **f** Output power density and energy conversion efficiency of the 3D SCOF membrane compared with the state-of-the-art membranes. Corresponding membranes were shown in Supplementary Table 4.

not very high energy conversion efficiency. Compared with 2D COF membranes, the synergetic effect of highly charged pore walls, sub-1 nm pore size, and three-dimensional interconnected pore channels for our 3D SCOF membrane increased both ion selectivity and enhanced ion flux, contributing to the maximal ion perms-electivity and thus substantially high power density and conversion efficiency.

## Discussion
In summary, we demonstrated the design and fabrication of 3D sulfonic acid-functionalized COF (3D SCOF) membranes for selective and efficient ion transport by using a dual acid-mediated interfacial polymerization approach. Through the controlled assembly of sulfonic acid-functionalized aldehyde monomers and tetra-(4-anilyl) methane monomers by the mediation of octanoic acid as the organic phase and

acetic acid in the aqueous phase, tight and defect-free 3D SCOF membranes were fabricated. Featuring the low-energy-barrier ion transport pathways and abundant sulfonate groups, the 3D SCOF membrane exhibited a high proton conductivity of 843 mS cm$^{-1}$ at 90°C. Besides, the highly interconnected ultramicroporous channels (0.97 nm) contribute to a concomitant increase in ion conductivity and ion selectivity. When utilized in osmotic energy conversion, the maximum output power density was 21.2 W m$^{-2}$, the ion selectivity was 0.976 and the energy conversion efficiency was 45.3% under a 50-fold salinity gradient between seawater and river water, outperforming most state-of-the-art membranes. This study not only develops a facile method to construct 3D ionic COF membranes but also builds a nascent platform of 3D ionic COF materials for energy-related applications.

## Methods

### Preparation of 3D SCOF membrane

The free-standing 3D SCOF membrane was prepared via the interfacial polymerization of tetra-(4-anilyl)-methane (monomer B) and sulfonic acid-functionalized aldehyde (monomer A). For AA/OA system preparation, for instance, monomer B and monomer A were dissolved in 30 mL octanoic acid (0.01 mmol/mL) and 10 mL acetic acid solution (0.01 mmol/mL, 7 mL water and 3 mL 6 M acetic acid), respectively. The resulting suspension was stirred, sonicated and filtered to form a clear and transparent solution. After that, the octanoic acid solution was added dropwise to the surface of acetic acid solution to construct reaction system. The organic-aqueous system remained undisturbed at room temperature for several days. After the reaction of several days, a self-standing thin 3D SCOF membrane was clearly observed at the organic/water interface. To collect the resultant membrane, the upper organic solution was removed by a pipette, leaving the membrane floating on the water surface. Subsequently, the membrane was dredged up by approaching it slowly and perpendicularly using a spatula or other substrates such as filter membranes, and then washed thoroughly with tetrahydrofuran, acetone, water and ethanol in turn. Owing to its flexibility and mechanical stability, the membrane kept intact after the transfer operation. The membrane was preserved in ethanol solvent for further characterizations and performance tests. Based on the AA/OA system, PTSA/OA, TFA/OA, AA/HA and AA/DA systems were also attempted to be applied to prepare the membranes following similar procedures. For single acid system preparation, the monomer A was dissolved in water with acetic acid and mesitylene as the organic phase.

### Characterization

The chemical structure of the 3D SCOF membrane was characterized by Fourier transform infrared (FTIR, BRUKER Vertex70) and the solid-state $^{13}$C CP-MAS NMR spectra on a Varian infinity plus 300 MHz spectrometer (12 kHz). The morphology of the 3D SCOF membranes was characterized by field-emission scanning electron microscope (SEM, Nanosem 430), transmission electron microscopy (TEM JEM-2100F, Japan) and atomic force microscopy (AFM Bruker Dimension Icon, USA). Powder X-ray diffraction (PXRD, Bruker D8, $\lambda = 1.5406$ Å, 8° min$^{-1}$) was utilized to investigate the structure of the 3D SCOF membrane. Thermal stability of the 3D SCOF membrane was measured by TGA-50 SHIMADZU from 40 °C to 800 °C with a ramp rate of 10 °C min$^{-1}$ under nitrogen atmosphere. The nitrogen adsorption and desorption isotherms were measured by a Micrometrics ASAP-2046 analyzer at 77 K.

## Data availability

All data supporting the findings of this study are available within the article and the Supplementary Information file or available from the corresponding authors upon request. Source data are provided with this paper.

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

## Acknowledgements

This work is financially supported by the National Natural Science Foundation of China, grant No. 21838008 (Z.J.), U20B2023 (Z.J.) and 21621004 (Y.Y.).

## Author contributions

Z.J., L.C., Y.K., and T.Z. conceived the idea and designed the research plan. T.Z., L.C., and Y.K. carried out the experiment. T.Z. and B.L. carried out Fukui function calculation, diffusion calculation and interaction energy calculations. H.W., B.S., X.W., X.P., C.F. and C.Y. provided constructive suggestions for results and discussion. All authors participated in the discussion. Y.K., T.Z., L.C., and Z.J. co-wrote the manuscript.

## Competing interests

The authors declare no competing interests.
