## [Peer Review File · Nature Communications]

3D Covalent Organic Framework Membrane with Fast and Selective Ion TransportReviewers' Comments:

Reviewer #1:

Remarks to the Author:

In this submission, the authors reported the fabrication of a 3D sulfonic acid-functionalized COF membrane using a dual acid-mediated interfacial polymerization approach, which was used for proton conduction and osmotic energy harvesting. This manuscript is suggested to be published in Nature Communication after addressing the following issues:

1. Numerous COF-based membranes have been developed for osmotic energy harvesting, which are suggested to be listed in Fig 3f and along this line, the advantage of 3D COF compared with 2D COFs is suggested to be highlighted.
2. There are great differences between the simulated XRD pattern and the experimental one. How the authors determine the pore size from the XRD pattern.
3. The Y-axis of the N₂ adsorption isotherm cannot be deleted, which is meaningful.
4. The equation for calculating the output power density is not correct, where the current should be current density.
5. The stability of the synthesized membrane is suggested to be evaluated.
6. Some figures are very vague and the format of reference is not consistent.

Reviewer #2:

Remarks to the Author:

High-performance ion-conductive membranes are highly desirable for electrochemical energy-related applications. This paper reported the design and fabrication of 3D sulfonic acid-functionalized COF (3D SCOF) membranes using a dual acid-mediated interfacial polymerization approach they developed. The resulting 3D SCOF membrane exhibited superior proton conductivity, particularly, very high ion selectivity and the energy conversion efficiency under a 50-fold salinity gradient. This is an interesting work with comprehensive and insightful studies warranting publication in Nature Communications. However, revisions are needed with the following minor comments addressed.

1. In Fig. 1, the authors showed a picture of the prepared membrane with a thickness of only 1 micrometer. Considering the difficulty in taking out and transferring the self-standing membrane from the container, the author should give more detailed description about the whole process in the Supplementary Materials.
2. There are no Nyquist plots in both main text or Supplementary Materials, and only the proton conductivity data are presented. Nyquist plots of 3D SCOF membrane corresponding to Fig.3 (a) should be provided.
3. It is mentioned that the resulting 3D SCOF membrane had extremely high proton conductivity, and the stable conduction is also one of the important characteristics. If possible, please add the relevant data about the conduction stability to the Supplementary Materials.
4. The calculation formula for energy conversion efficiency is missing in both the main texts and Supplementary Materials. More calculation details should be incorporated into Supplementary Materials.
5. The format of the tables in the Supplementary Materials should be adjusted to be consistent in all places.
6. Some closely related literatures are suggested to be cited: ACS Energy Lett. 2022, 7, 2937-2943; EnergyChem, 2022, 4, 100079; Adv. Funct. Mater. 2022, 32, 2109210.

Reviewer #3:

Remarks to the Author:

This manuscript titled in "3D Covalent Organic Framework Membrane with Fast and Selective Ion Transport" reported the fabrication of a 3D sulfonic acid-functionalized COF membrane for efficient

and selective ion transport, using dual acid-mediated interfacial polymerization strategy. The 3D SCOF achieved the high power density and efficiency of osmotic energy conversion. Before considering this manuscript for publication in Nature Communications, the authors should consider the following points in any revision follows:

1. The authors should give the high-resolution TEM photographs to verify the crystalline lattice rather than photo as Fig. 1g.
2. The authors should give show the devices of proton conductivity and osmotic energy conversion testing.
3. The authors should write the kind electrodes, which were applied in proton conductivity and osmotic energy conversion performance test.
4. The authors said "more than 4-fold larger than the commercialization benchmark", the stability of membrane in osmotic energy conversion should be measured.
5. The effective area of testing with osmotic energy conversion should be presented.
6. Photograph of a 3D SCOF membrane (soaked in water) (Fig. 1d). Can the membrane be picked up or is it only in the water? The author should give evidence.
7. The typical stress-strain curve of membrane should be measured.

Response to reviewers

Reviewer #1 (Remarks to the Author):

In this submission, the authors reported the fabrication of a 3D sulfonic acid-functionalized COF membrane using a dual acid-mediated interfacial polymerization approach, which was used for proton conduction and osmotic energy harvesting. This manuscript is suggested to be published in Nature Communication after addressing the following issues:

Thank the reviewer for the highly positive remarks and valuable guidance on our manuscript.

1. Numerous COF-based membranes have been developed for osmotic energy harvesting, which are suggested to be listed in Fig 3f and along this line, the advantage of 3D COF compared with 2D COFs is suggested to be highlighted.

Reply: Thank the reviewer for the valuable guidance.

The osmotic energy conversion performance of reported COF-based membranes has been added to the revised figure, and the related descriptions about the comparison between 2D COF and 3D COF membrane have been added to the revised main text.

The comparison with previously reported membranes is shown in Fig. R1 in terms of output power density and energy conversion efficiency. Our 3D SCOF membrane shows a superior energy conversion efficiency of 45.12%, which is the highest among all existing materials utilized in osmotic energy conversion, with a simultaneously high output power density of 22.2 W m^{-2} , outperforming most of state-of-the-art

membranes. It should be noted for 2D COF membrane, the abundant ordered channels of 2D COF membrane are conducive to ion diffusion for achieving high power density, but the relatively larger pore sizes than 1 nm generate inadequate ion selectivity and thus not very high energy conversion efficiency. Compared with 2D COF membranes, the synergetic effect of highly charged pore walls, sub-1 nm pore size, and three-dimensional interconnected pore channels for our 3D SCOF membrane increases both ion selectivity and ion flux, contributing to the maximal ion permselectivity and thus substantially high power density and conversion efficiency.

The original Fig. 3f has been revised to

Fig. R1 Output power density and energy conversion efficiency of the 3D SCOF membrane compared with the state-of-the-art membranes. Corresponding membranes were shown in Supplementary Table 4.

The corresponding Supplementary Table 4 has been revised to

Supplementary Table 4. The osmotic energy conversion performance of previously reported membranes in Fig. 3f.

Materials	Channel size (nm)	Power density (W^{-2} , NaCl)	Energy conversion efficiency (%)	Ref.
SPX membrane	0.5-0.7	1.23 (50-fold)	37.1	1
Ti ₃ C ₂ T _x Mxene	0.64	21.00 (1000-fold)	40.6	2
GO/SNF/GO	0.75	5.07 (50-fold)	14.6	3
SPEEK membrane	2.7	5.80 (50-fold)	16.7	4
CMWs	3.5	2.78 (50-fold)	36.3	5
Hydrogel membrane	7	5.38 (50-fold)	14.5	6
Janus membrane	M-1:10/M-2:17	2.10 (50-fold)	17.5	7
ABN30 membrane	-	7.70 (50-fold)	5.9	8
MOF/SPEEK	-	7.00 (50-fold)	40.0	9
Vertical-aligned BN nanopores	30	106 (1000-fold)	12.0	10
Vertical-aligned GO	0.86	10.60 (50-fold)	18.7	11
Vertical-aligned Mxene	0.68-1.07	4.6 (50-fold)	44.2	12
BDA-TAM COF	1.4	5.31 (KCL, 50-fold)	28.3	13
TFP-TPA COF@ANM	1.1	27.8 (500-fold)	45.3	14
TpEB@TpPa-SO ₃ Na	1-2	19.2 (50-fold)	26.9	15
3D SCOF membrane	0.97	22.2 (50-fold)	45.1	This Work!

“Overall, the synergetic effect of highly charged pore walls, sub-1 nm pore size, and three-dimensional interconnected pore channels rendered the increased ion selectivity and enhanced ion flux, contributing to the maximal ion permselectivity and thus substantially high power density and conversion efficiency.” has been revised to “The comparison with previously reported membranes was shown in Fig. 3f from output power density and energy conversion efficiency. Our 3D SCOF membrane

showed a superior energy conversion efficiency of 45.12%, which was the highest among all existing materials utilized in osmotic energy conversion, with a simultaneously high output power density of 22.2 W m⁻², outperforming most of state-of-the-art membranes. It should be noted for 2D COF membrane, the abundant ordered channels of 2D COF membrane were conducive to ion diffusion for achieving high power density, but the relatively larger pore sizes than 1 nm generated inadequate ion selectivity and thus not very high energy conversion efficiency. Compared with 2D COF membranes, the synergetic effect of highly charged pore walls, sub-1 nm pore size, and three-dimensional interconnected pore channels for our 3D SCOF membrane increased both ion selectivity and enhanced ion flux, contributing to the maximal ion permselectivity and thus substantially high power density and conversion efficiency.”

2. There are great differences between the simulated XRD pattern and the experimental one. How the authors determine the pore size from the XRD pattern.

Reply: Thank the reviewer for the valuable guidance. The XRD pattern of the simulated 3-fold interpenetrated diamond topology coincides with the experimental result with the relatively good agreement factors of $R_{wp} = 10.49\%$ and $R_p = 8.09\%$. The differences between the simulated and experimental XRD patterns around 20° may be ascribed to the residual reaction monomers in the framework. Therefore, the 3D SCOF is proposed to adopt a 3-fold interpenetrated dia framework and the pore size of 3D SCOF membrane is predicted to be 0.97 nm based on the diameter of the largest included sphere in the framework. To confirm the channel size of the 3D

SCOF membrane, the rejection tests of polyethylene glycols (PEG) with various molecular weights (600, 1500, 2000, and 4000 Da) are conducted on a dead-end filtration device. The average pore size of 3D SCOF membrane is defined as the same as the geometric mean diameter of the PEG with 50% rejection¹⁶. For PEG, the Stokes radii of PEG can be calculated from the equation $r = 16.73 \times 10^{-12} \times M_w^{0.557}$. As shown in Fig. R2, the molecular weight of 3D SCOF membrane based on the 50% rejection of PEG is 1490 Da and therefore the mean channel diameter is about 0.98 nm, which is consistent with the predicted value.

Fig. R2 Molecular rejection of PEG by the 3D SCOF membrane as a function of molecular weight in the water system.

Fig. R2 (Supplementary Fig. 8) has been added to the Supplementary Materials.

“The pore size distribution analysis was calculated by using the nonlocal density functional theory (NLDFE), demonstrating a mean pore width of 0.97 nm for the 3D SCOF membrane.” has been revised to “The pore size of 3D SCOF membrane was predicted to be 0.97 nm based on the diameter of the largest included sphere in the

framework. To confirm the channel size of the 3D SCOF membrane, the rejection of polyethylene glycols (PEG) with various molecular weights were determined on a dead-end filtration device. As shown in Supplementary Fig. 8, the mean channel diameter of the 3D SCOF membrane calculated based on PEG rejection was about 0.98 nm, which was consistent with the predicted value.”

“The rejection tests of 3D SCOF membranes were conducted on a homemade dead-end filtration device at 2 bar based on polyethylene glycols (PEG) with various molecular weights (600, 1500, 2000, and 4000 Da). Before the rejection test, each PEG was dissolved in DI water to form a feed solution with a concentration of 200 ppm. Concentrations of PEGs in the feed and permeate solution were detected by total organic carbon (TOC) analyzer. The rejection rates (R) were calculated by the following equation:

$$R = \left(1 - \frac{c_f}{c_p}\right) \times 100\%$$

where c_f and c_p are the solution concentration in the feed and permeate solution, respectively.

The average pore size was defined as the same as the geometric mean diameter of the PEG with 50% rejection. The Stokes radii of PEG can be calculated from the following equation: $r = 16.73 \times 10^{-12} \times M_w^{0.557}$.” has been added to “**Pore size distribution tests**” section of the Supplementary Materials.

3. The Y-axis of the N₂ adsorption isotherm cannot be deleted, which is meaningful.

Reply: Thank the reviewer for the valuable guidance. The Y-axis of the nitrogen adsorption-desorption isotherm has been supplemented on the figure.

The original Fig. S6 has been revised to

Fig. R3 (Supplementary Fig. 7). N₂ adsorption-desorption isotherms at 77K of 3D SCOF membrane.

4. The equation for calculating the output power density is not correct, where the current should be current density.

Reply: Thank the reviewer for the valuable guidance. The equation for calculating the output power density in Supplementary Materials has been corrected.

The original description “The osmotic power output was calculated using the following equation: $P_{output} = I^2 \times R$, Where, I is the recorded current, and R is the external load resistance.” has been revised to “The output power (P_L) and output power density (PD) were calculated using the following equations:

$$P_L = I^2 \times R_L$$

$$PD = P_L/S$$

where I is the current, R_L is the external load resistance, and S is the effective testing area of the membrane.”

5. The stability of the synthesized membrane is suggested to be evaluated.

Reply: According to the reviewer's guidance, the long-term stability of 3D SCOF membrane in proton conduction and osmotic energy conversion has been measured and the related descriptions have been added to the revised main text and Supplementary Materials. As shown in Fig. R4, the proton conduction stability test of the 3D SCOF membrane was performed at 60°C, 100% RH for a week. The proton conductivity of the membrane remained almost unchanged, which demonstrated the stable structure of the membrane. Fig. R5 further demonstrated the excellent long-term stability of the membrane in osmotic energy conversion. A negligible decrease was observed in the measured power density after 6 days, which can be ascribed to the good chemical and physical structural stability of the 3D SCOF membrane. After the long-term stability test, the membrane was still very robust and there were no obvious morphology changes. These observations clearly indicated the great application viability of the 3D SCOF membrane.

Fig. R4 The long-term stability of 3D SCOF membrane in proton conduction at 60°C, 100% RH.

Fig. R5 The long-term stability of 3D SCOF membrane in osmotic energy conversion under the standard artificial river water and sea water system.

“In addition, we evaluated the proton conduction stability test of the 3D SCOF membrane at 60°C, 100% RH for a week (Supplementary Fig. 22). The proton conductivity of the membrane remained almost unchanged after several days, which demonstrated the stable structure of the membrane.” has been added to the main text (page 11, line 239).

“Furthermore, we evaluated the long-term stability of the membrane in osmotic energy conversion at a 50-fold concentration gradient (Supplementary Fig. 24). A negligible decrease was observed in the measured power density after 6 days, which can be ascribed to the good chemical and physical structural stability of the 3D SCOF membrane. The observation clearly demonstrated the great application reliability of the 3D SCOF membrane.” has been added to the main text (page 13, line 274).

“For the long-term stability test, the membrane was clamped between the two electrodes and stayed in the testing conditions (60°C, 100% RH) all the time.” has been added to “**Proton conductivity measurement**” section of the Supplementary

Materials.

“For the long-term stability test, the membrane was immersed in the testing solution all the time, and the testing solutions were refreshed before each measurement.” has been added to “**Osmotic energy conversion**” section of the Supplementary Materials.

Fig. R4 (Supplementary Fig. 22) and R5 (Supplementary Fig. 24) have been added to the Supplementary Materials.

6. Some figures are very vague and the format of reference is not consistent.

Reply: We are sorry for that figures in the main text and Supplementary Materials are not clear enough and the format of reference are not consistent. The related figures have been replaced with clearer ones and the format of references has been unified into the required format.

References:

1. Zhu, Q. *et al.* A sulfonated ultramicroporous membrane with selective ion transport enables osmotic energy extraction from multiform salt solutions with exceptional efficiency. *Energy Environment Science* **15**, 4148–4156 (2022).
2. Hong, S. *et al.* Two-Dimensional $\text{Ti}_3\text{C}_2\text{T}_x$ Mxene Membranes as Nanofluidic Osmotic Power Generators. *ACS Nano* **13**, 8917–8925 (2019).
3. Xin, W. *et al.* Biomimetic Nacre-Like Silk-Crosslinked Membranes for Osmotic Energy Harvesting. *ACS Nano* **14**, 9701–9710 (2020).
4. Zhao, Y. *et al.* Robust sulfonated poly (ether ether ketone) nanochannels for high-performance osmotic energy conversion. *National Science Review* **7**, 1349–1359 (2020).
5. Xie, L. *et al.* Sequential Superassembly of Nanofiber Arrays to Carbonaceous

- Ordered Mesoporous Nanowires and Their Heterostructure Membranes for Osmotic Energy Conversion. *Journal of the American Chemical Society* **143**, 6922–6932 (2021).
6. Chen, W. *et al.* Improved Ion Transport and High Energy Conversion through Hydrogel Membrane with 3D Interconnected Nanopores. *Nano Letters* **20**, 5705–5713 (2020).
 7. Zhang, Z. *et al.* Ultrathin and Ion-Selective Janus Membranes for High-Performance Osmotic Energy Conversion. *J Journal of the American Chemical Society* **139**, 8905–8914 (2017).
 8. Liu, J. *et al.* Self-standing and flexible covalent organic framework (COF) membranes for molecular separation. *Science Advances* **6**, eabb1110 (2020).
 9. Zhao, X. *et al.* Metal organic framework enhanced SPEEK/SPSF heterogeneous membrane for ion transport and energy conversion. *Nano Energy* **81**, 105657 (2021).
 10. Pendse, A. *et al.* Highly Efficient Osmotic Energy Harvesting in Charged Boron-Nitride-Nanopore Membranes. *Advanced Functional Materials* **31**, 2009586 (2021).
 11. Zhang, Z. *et al.* Vertically Transported Graphene Oxide for High-Performance Osmotic Energy Conversion. *Science Advances* **7**, 2000286 (2020).
 12. Ding, L. *et al.* Oppositely Charged $Ti_3C_2T_x$ Mxene Membranes with 2D Nanofluidic Channels for Osmotic Energy Harvesting. *Angewandte Chemie - International Edition* **59**, 8720–8726 (2020).
 13. Wang, C. *et al.* Ultrathin Self-Standing Covalent Organic Frameworks toward Highly-Efficient Nanofluidic Osmotic Energy Generator. *Advanced Functional Materials* **32**, 2204068 (2022).
 14. Gao, M. *et al.* A bioinspired ionic diode membrane based on sub-2 nm covalent organic framework channels for ultrahigh osmotic energy generation. *Nano Energy* **105**, 108007 (2023).
 15. Cao, L. *et al.* An Ionic Diode Covalent Organic Framework Membrane for Efficient Osmotic Energy Conversion. *ACS Nano* **16**, 18910–18920 (2022).

16. Gao, J. et al. Polyethyleneimine (PEI) cross-linked P84 nanofiltration (NF) hollow fiber membranes for Pb^{2+} removal. *Journal of Membrane Science* **452**, 300–310 (2014).

Reviewer #2 (Remarks to the Author):

High-performance ion-conductive membranes are highly desirable for electrochemical energy-related applications. This paper reported the design and fabrication of 3D sulfonic acid-functionalized COF (3D SCOF) membranes using a dual acid-mediated interfacial polymerization approach they developed. The resulting 3D SCOF membrane exhibited superior proton conductivity, particularly, very high ion selectivity and the energy conversion efficiency under a 50-fold salinity gradient. This is an interesting work with comprehensive and insightful studies warranting publication in Nature Communications. However, revisions are needed with the following minor comments addressed.

Thank the reviewer for the highly positive remarks and valuable guidance on our manuscript.

1. In Fig. 1, the authors showed a picture of the prepared membrane with a thickness of only 1 micrometer. Considering the difficulty in taking out and transferring the self-standing membrane from the container, the author should give more detailed description about the whole process in the Supplementary Materials.

Reply: According to the reviewer's valuable guidance, the detailed description of the operation method for taking out and transferring the membrane from the interfacial system has been supplemented in the main text and Supplementary Materials.

“Preparation of 3D SCOF membrane” section of the Methods in the main text:

The original description “The membrane (3D SCOFM-AA/OA) was then collected from the interface for later characterization and testing.” has been revised to “After

the reaction of several days, a self-standing thin 3D SCOF membrane was clearly observed at the organic/water interface. To collect the resultant membrane, the upper organic solution was removed by a pipette, leaving the membrane floating on the water surface. Subsequently, the membrane was dredged up by approaching it slowly and perpendicularly using a spatula or other substrates such as filter membranes, and then washed thoroughly with tetrahydrofuran, acetone, water and ethanol in turn. Owing to its flexibility and mechanical stability, the membrane kept intact after the transfer operation. The membrane was preserved in ethanol solvent for the subsequent characterizations and performance tests.”

Fig. R6 Photographs of (a) the interfacial reaction system, (b) the 3D SCOF membrane on the water solution surface, (c) on a spatula, and (d) in ethanol solvent.

Fig. R6 (Supplementary Fig. 27) has been added to the Supplementary Materials.

2. There are no Nyquist plots in both main text or Supplementary Materials, and only the proton conductivity data are presented. Nyquist plots of 3D SCOF membrane corresponding to Fig.3 (a) should be provided.

Reply: According to the reviewer’s valuable guidance, the Nyquist plots of the 3D SCOF membrane corresponding to Fig. 3a have been supplemented in the revised Supplementary Materials as shown below:

Fig. R7 Nyquist plots of 3D SCOF membrane measured at different temperatures under 100% RH.

The original description “The proton conductivity of the 3D SCOF membrane increased with the temperature and achieved a maximum proton conductivity of 843 mS cm^{-1} at 90°C, 100% RH.” has been revised to “The proton conductivity of the 3D SCOF membrane increased with the temperature and achieved a maximum proton conductivity of 843 mS cm^{-1} at 90°C, 100% RH (Fig. 3a and Supplementary Fig. 20)”

Fig. R7 (Supplementary Fig. 20) has been added to the Supplementary Materials.

3. It is mentioned that the resulting 3D SCOF membrane had extremely high proton conductivity, and the stable conduction is also one of the important characteristics. If possible, please add the relevant data about the conduction stability to the Supplementary Materials.

Reply: According to the reviewer’s guidance, the long-term stability of 3D SCOF membrane in proton conduction has been measured and the related descriptions have been added to the revised main text and Supplementary Materials. As shown in Fig. R8, the proton conduction stability test of the 3D SCOF membrane was performed at 60°C, 100% RH for a week. The proton conductivity of the membrane remained almost unchanged, which demonstrated the stable chemical and physical structure of the membrane.

Fig. R8 The long-term stability of 3D SCOF membrane in proton conduction at 60°C, 100% RH.

“In addition, we evaluated the proton conduction stability test of the 3D SCOF membrane at 60°C, 100% RH for a week (Supplementary Fig. 22). The proton conductivity of the membrane remained almost unchanged, which demonstrated the stable structure of the membrane.” has been added to the main text (page 11, line 239).

“For the long-term stability test, the membrane was clamped between the two electrodes and stayed in the testing conditions (60°C, 100% RH) all the time.” has

been added to “**Proton conductivity measurement**” section of the Supplementary Materials.

Fig. R8 (Supplementary Fig. 22) has been added to the Supplementary Materials.

4. The calculation formula for energy conversion efficiency is missing in both the main texts and Supplementary Materials. More calculation details should be incorporated into Supplementary Materials.

Reply: According to the reviewer’s guidance, the corresponding calculation formula and detailed descriptions have been added to the Supplementary Materials.

“**Energy conversion efficiency**” section in Supplementary Materials:

$$E_{diff} = V_{oc} - E_{redox}$$

$$E_{redox} = \frac{RT}{zF} \ln \frac{\gamma_{C_H} C_H}{\gamma_{C_L} C_L}$$

$$t = \frac{1}{2} \left(\frac{E_{diff}}{\frac{RT}{zF} \ln \left(\frac{\gamma_{C_H} C_H}{\gamma_{C_L} C_L} \right)} + 1 \right)$$

Where γ and c represent activity coefficient and concentration of ions; R , T , F and z refer to the universal gas constant, absolute temperature, Faraday constant, and charge number, respectively.” has been revised to “The diffusion potential (E_{diff}) can be calculated as:

$$E_{diff} = V_{oc} - E_{redox}$$

$$E_{redox} = \frac{RT}{zF} \ln \frac{\gamma_{C_H} C_H}{\gamma_{C_L} C_L}$$

where V_{oc} , E_{redox} , R , T , F , z , γ and c refer to the open-circuit voltage, redox potential, universal gas constant, absolute temperature, Faraday constant, charge number, activity coefficient of ions, and ion concentration, respectively.

For a given concentration gradient, the cation transference number (t) can be

calculated as:

$$t = \frac{1}{2} \left(\frac{E_{diff}}{\frac{RT}{zF} \ln\left(\frac{Y_{CH}C_H}{Y_{CL}C_L}\right)} + 1 \right)$$

Accordingly, the maximum energy conversion efficiency (η_{max}) can be calculated as:

$$\eta_{max} = \frac{1}{2}(2t - 1)^2$$

When the membrane is ideally cation selective, t reaches the max of 1, and the maximum efficiency reaches the max of 50%.”

5. The format of the tables in the Supplementary Materials should be adjusted to be consistent in all places.

Reply: According to the reviewer’s guidance, the format of related tables in the Supplementary Materials has been unified into the required format.

6. Some closely related literatures are suggested to be cited: ACS Energy Lett. 2022, 7, 2937-2943; Energy Chem, 2022, 4, 100079; Adv. Funct. Mater. 2022, 32, 2109210.

Reply: Thank the reviewer for the valuable guidance, all the above references have been added to the main text.

“In this respect, the porous membrane with tunable pore structure and designable channel surface is a promising platform to produce advanced membranes¹.”

“As an intriguing type of porous crystalline material, ionic covalent organic framework (COF) membranes have displayed huge potential as high-performance ion conductors due to the permanently ordered channels and precise arrangement of ionic groups²⁻³.”

1. Song, Y. *et al.* Advanced porous organic polymer membranes: Design, fabrication,

and energy-saving applications. *EnergyChem* **4**, 100079 (2022).

2. Zhu, C. *et al.* Integration of Thermoelectric Conversion with Reverse Electrodialysis for Mitigating Ion Concentration Polarization and Achieving Enhanced Output Power Density. *ACS Energy Letters* **7**, 2937-2943 (2022).
3. Zhu, C. *et al.* Manipulating Charge Density in Nanofluidic Membranes for Optimal Osmotic Energy Production Density. *Advanced Functional Materials* **32**, 2109210 (2022)

Reviewer #3 (Remarks to the Author):

This manuscript titled in “3D Covalent Organic Framework Membrane with Fast and Selective Ion Transport” reported the fabrication of a 3D sulfonic acid-functionalized COF membrane for efficient and selective ion transport, using dual acid-mediated interfacial polymerization strategy. The 3D SCOF achieved the high power density and efficiency of osmotic energy conversion. Before considering this manuscript for publication in Nature Communications, the authors should consider the following points in any revision follows:

Thank the reviewer for the highly positive remarks and valuable guidance on our manuscript.

1. The authors should give the high-resolution TEM photographs to verify the crystalline lattice rather than photo as Fig. 1g.

Reply: According to the reviewer’s guidance, the high-resolution TEM image has been presented in the revised main text. As shown in Fig. R9, the lattice patterns in different orientations are clearly observed all over the visual field, illustrating the high crystallinity of the 3D SCOF membrane. Particularly, the zonary bright regions reveal the highly ordered pores of the 3D framework structure in the membrane.

Fig. R9 The high-resolution TEM image of the 3D SCO membrane.

Fig. R9 has been added in the main text as Fig. 1h.

“In addition, the high crystallinity of the 3D SCO membrane was also verified by the lattice patterns all over the visual field in the high-resolution TEM image (Fig. 1h). Particularly, the zony bright regions revealed the highly ordered pores of the 3D framework structure in the membrane.” has been added to the main text (page 7, line 142).

2. The authors should give show the devices of proton conductivity and osmotic energy conversion testing.

Reply: According to the reviewer’s guidance, the photographs of device for proton conductivity and osmotic energy conversion tests have been shown in the revised Supplementary Materials.

“**Proton conductivity measurement**” section in Supplementary Materials:

Fig. R10 Schematic illustration of the experimental setup device for proton conductivity measurement.

“Osmotic energy conversion” section in Supplementary Materials:

Fig. R11 Photograph of the experimental setup device for osmotic energy conversion performance test. The device is connected to the external detection equipment for collecting output current and voltage.

In the main text, “To evaluate the ion transport performance of the 3D SCOF membrane (3D SCOFM-AA/OA), the proton conductivity was acquired by using electrochemical impedance spectroscopy.” has been revised to “To evaluate the ion

transport performance of the 3D SCOF membrane (3D SCOFM-AA/OA), the proton conductivity was acquired by using electrochemical impedance spectroscopy based on an electrochemical workstation (Supplementary Fig. 19).”

In the main text, “We further investigated the osmotic energy conversion capability of the 3D SCOF membrane (Fig. 3c).” has been revised to “We further investigated the osmotic energy conversion capability of the 3D SCOF membrane (Fig. 3c and Supplementary Fig. 23).”

Fig. R10 (Supplementary Fig. 19) and R11 (Supplementary Fig. 23) have been added to the Supplementary Materials.

3. The authors should write the kind electrodes, which were applied in proton conductivity and osmotic energy conversion performance test.

Reply: Thank the reviewer for the guidance. For the proton conductivity measurement, we use two silver electrodes to reduce the electrode polarization. For the osmotic energy conversion test, a pair of fresh Ag/AgCl electrodes (5 mm × 20 mm × 0.2 mm) are used to apply a transmembrane electrical potential.

“The membranes were cut into strip samples of 0.3-0.5 cm wide and clamped between the electrodes.” has been revised to “The membrane was cut into strip sample of 0.3-0.5 cm wide and clamped across two silver electrodes spaced 1 cm apart.”

“A pair of fresh Ag/AgCl electrodes (5 mm × 20 mm × 0.2 mm) was embedded in the two side chambers and then sealed with epoxy resin and polyimide tape.” has been revised to “The 3D SCOF membranes were mounted in between a two-chamber cell wherein one chamber contained artificial river water (0.01 M NaCl solution) and the

other contained artificial seawater (0.5 M NaCl solution). A pair of fresh Ag/AgCl electrodes (5 mm × 20 mm × 0.2 mm) were embedded in the two side chambers. For the osmotic power output, the cell was connected to an external load resistor (MC-21-B, Mingcheng, Shenzhen, China) and the current across the resistance was recorded using a source meter (KEITHLEY, 2450 SourceMeter®). ”

4. The authors said “more than 4-fold larger than the commercialization benchmark”, the stability of membrane in osmotic energy conversion should be measured.

Reply: According to the reviewer’s valuable guidance, a long-term stability test under the standard artificial river water and sea water system is performed for 6 days to evaluate the stability of the 3D SCOF membrane in osmotic energy conversion. As shown in Fig. R12, a negligible decrease is observed in the measured power density after 6 days, which can be ascribed to the good chemical and physical structural stability of the 3D SCOF membrane. After the long-term stability test, the membrane is still very robust and there are no observable morphology changes. These observations clearly indicate the great application viability of the 3D SCOF membrane.

Fig. R12 The long-term stability of 3D SCOF membrane in osmotic energy conversion under the standard artificial river water and sea water system.

“Furthermore, we evaluated the long-term stability of the membrane in osmotic energy conversion at a 50-fold concentration gradient (Supplementary Fig. 24). A negligible decrease was observed in the measured power density after 6 days, which can be ascribed to the good chemical and physical structural stability of the 3D SCOF membrane. The observation clearly demonstrated the great application reliability of the 3D SCOF membrane.” has been added to the main text (page 13, line 274).

“For the long-term stability test, the membrane was immersed in the testing solution all the time, and the testing solutions were refreshed before each measurement.” has been added to “**Osmotic energy conversion**” section of the Supplementary Materials.

Fig. R12 (Supplementary Fig. 24) has been added to the Supplementary Materials.

5. The effective area of testing with osmotic energy conversion should be presented.

Reply: Thank the reviewer for the valuable guidance. The effective testing area of 3D SCOF membrane is 0.03 mm², the same as previous reports.

“It should be noted that the effective testing area of 3D SCOF membrane was 0.03 mm², the same as previous reports.” has been added to the main text (page 12, line 263).

6. Photograph of a 3D SCOF membrane (soaked in water) (Fig. 1d). Can the membrane be picked up or is it only in the water? The author should give evidence.

Reply: Thank the reviewer for the valuable guidance. The self-standing 3D SCOF

membrane with high flexibility is about 1 micrometer thick and therefore is difficult to be picked up alone, but the membrane can be transferred to other substrates such as silicon wafer and filter membranes. Moreover, the membrane can be preserved in both wet and dry state. As shown in Fig. R13, we transfer the membrane soaked in water to a polyacrylonitrile substrate and dry the membrane in a fume hood for 24 hours. It can be observed that the membrane is still intact and there are no observable morphology changes.

Fig. R13. Photographs of the 3D SCOF membrane (a) on a spatula, (b) soaked in water and on a polyacrylonitrile substrate (c) before and (d) after drying. It can be observed that the membrane was still intact and there were no observable morphology changes.

“The membrane can be preserved in both wet and dry state (Supplementary Fig. 4).”
has been added to the main text (page 5, line 104).

Fig. R13 (Supplementary Fig. 4) has been added to the Supplementary Materials.

7. The typical stress-strain curve of membrane should be measured.

Reply: Thank the reviewer for the valuable guidance. The self-standing 3D SCOF membrane is about 1 micrometer thick and therefore is difficult to be picked up alone and installed on the electronic tensile machine for mechanical test. Therefore, other methods are tried to demonstrate the mechanical robustness of the membrane. As shown in Fig. R14, the membrane obtained at the organic/aqueous interface remains intact after being pressed with a spoon, confirming its good flexibility and mechanical robustness. We also use AFM tip to determine the Young's modulus of the membrane (Fig. R15). The 3D SCOF membrane possesses a high Young's modulus of approximately 2.4 GPa on average.

Fig. R14 Photograph of the 3D SCOF membrane pressed with a spoon, showing the mechanical robustness of the membrane.

Fig. R15 Young's modulus of the 3D SCOF membrane tested using the peak force quantitative nanomechanical property mapping method. The 3D SCOF membrane possessed a high Young's modulus of approximately 2.4 GPa on average.

Fig. R14 (Supplementary Fig. 2) and R15 (Supplementary Fig. 3) have been added to the Supplementary Materials.

“As shown in Fig. 1e-g, the 3D SCOF membrane was intact, flexible and defect-free.” has been revised to “The as-prepared 3D SCOF membrane was intact, defect-free, flexible and mechanically robust (Figure 1e-f, Supplementary Fig. 2). As shown in Supplementary Fig. 3, the 3D SCOF membrane possessed a high Young's modulus of approximately 2.4 GPa on average.”

Reviewers' Comments:

Reviewer #1:

Remarks to the Author:

The authors carefully revised the manuscript. I appreciate the efforts of the authors for the clarification of the points raised by the reviewers. I am happy to recommend the acceptance of this manuscript in Nature Communications as is.

Reviewer #2:

Remarks to the Author:

The authors have satisfactorily addressed all the comments from the reviewer and the manuscript can now be accepted in its current form.

Reviewer #3:

Remarks to the Author:

[Note: Please see attached pdf]

Thanks for your serious responses. In this review I found a very serious problem, the authors' experimental results exceeded the theoretical values, which is not normal. In figure 3d and , the V_{oc} of system was measured 200 mV with 50-fold gradient of NaCl solution, that was not possible.

“**Energy conversion efficiency**” section in Supplementary Materials:

$$E_{diff} = V_{oc} - E_{redox}$$

$$E_{redox} = \frac{RT}{zF} \ln \frac{\gamma_{cH} c_H}{\gamma_{cL} c_L}$$

$$t = \frac{1}{2} \left(\frac{E_{diff}}{\frac{RT}{zF} \ln \left(\frac{\gamma_{cH} c_H}{\gamma_{cL} c_L} \right)} + 1 \right)$$

Where γ and c represent activity coefficient and concentration of ions; R , T , F and z refer to the universal gas constant, absolute temperature, Faraday constant, and charge number, respectively.” has been revised to “The diffusion potential (E_{diff}) can be calculated as:

$$E_{diff} = V_{oc} - E_{redox}$$

$$E_{redox} = \frac{RT}{zF} \ln \frac{\gamma_{cH} c_H}{\gamma_{cL} c_L}$$

where V_{oc} , E_{redox} , R , T , F , z , γ and c refer to the open-circuit voltage, redox potential, universal gas constant, absolute temperature, Faraday constant, charge number, activity coefficient of ions, and ion concentration, respectively.

From the calculation formula for energy conversion, the maximum value of V_{oc} could be calculated:

$$V_{OC_{MAX}} = 2 * E_{redox} = 2 \times \frac{8.3145 \times 298.15}{96485} \times \ln \frac{0.5 \times 0.681}{0.01 \times 0.903} = 186 \text{ mV} < 200 \text{ mV (the}$$

authors measured value)

The equation requires " $t = 1$ " for it to hold, which is impossible. As authors said "Furthermore, the corresponding cation selectivity (quantitatively described by cation transference number, t) was calculated to be 0.975, which was so close to the ideal cation selectivity ($t = 1$).” This is inconsistent with the experimental results.

We recommend that the editors consider the manuscript carefully for publication.

Response to reviewers

Reviewer (Remarks to the Author):

Thanks for your serious responses. In this review I found a very serious problem, the authors' experimental results exceeded the theoretical values, which is not normal. In figure 3d, the V_{OC} of system was measured 200 mV with 50-fold gradient of NaCl solution, that was not possible.

From the calculation formula for energy conversion, the maximum value of V_{OC} could be calculated: $V_{OC_{MAX}} = 2 \times E_{redox} = 2 \times \frac{8.3145 \times 298.15}{96485} \ln \frac{0.5 \times 0.681}{0.01 \times 0.903} = 186 \text{ mV} < 200 \text{ mV}$ (the authors measured value)

The equation requires " $t = 1$ " for it to hold, which is impossible. As authors said "Furthermore, the corresponding cation selectivity (quantitatively described by cation transference number, t) was calculated to be 0.975, which was so close to the ideal cation selectivity ($t = 1$).\" This is inconsistent with the experimental results.

We recommend that the editors consider the manuscript carefully for publication.

Reply: Thank the reviewer for the rigorous analysis, straightforward comments and very helpful suggestions.

Based on the reviewer's valuable guidance, the testing solutions of sodium chloride with different concentrations were re-prepared. Moreover, to minimize any influence from unexpected factors, we implemented an agarose salt bridge to connect the electrodes to the cell, which can eliminate the redox potential of the electrodes. In such way, the diffusion potential (E_{diff}) could be directly obtained from the intercept on the voltage axe of the current-voltage (I-V) curves. We re-conducted the current-voltage

(I-V) measurements in the two-compartment electrochemical cell connected with a salt bridge under the fresh 0.01 M NaCl and 0.5 M NaCl solution system. As shown in Fig. R1, the E_{diff} value is 88.9 mV and the corresponding cation selectivity (quantitatively described by cation transference number, t) is calculated to be 0.976.

$$t = \frac{1}{2} \left(\frac{E_{diff}}{\frac{RT}{zF} \ln \left(\frac{\gamma_{CH} C_H}{\gamma_{CL} C_L} \right)} + 1 \right) = \frac{1}{2} \left(\frac{0.0889}{\frac{8.3145 \times 298.15}{96485} \ln \left(\frac{0.681 \times 0.5}{0.903 \times 0.01} \right)} + 1 \right) = 0.976$$

The corresponding energy conversion efficiency is calculated to be 45.3%, which is the highest value reported.

$$\eta_{max} = \frac{1}{2} (2t - 1)^2 = \frac{1}{2} \times (2 \times 0.976 - 1)^2 = 45.3\%$$

Fig. R1 I-V curve of the 3D SCOF membrane under a 0.01 M/0.5 M NaCl concentration gradient.

In addition, we also re-tested the output power densities at the 5-fold, 50-fold, and 500-fold concentration gradient based on the fresh NaCl solutions (Fig. R2-R4). The corresponding maximum power densities achieve $\sim 4.8 \text{ W m}^{-2}$, 21.2 W m^{-2} and 69.6 W m^{-2} respectively, which are similar to those in our manuscript. These results verify the

superiority of our 3D SCOF membrane and the key conclusion of our work.

Fig. R2 Current density and power density of 3D SCOF membrane as functions of external resistances under a 5-fold concentration gradient.

Fig. R3 Current density and power density of 3D SCOF membrane as functions of external resistances under a 50-fold concentration gradient.

Fig. R4 Current density and power density of 3D SCOF membrane as functions of external resistances under a 500-fold concentration gradient.

The corresponding manuscript and Supplementary Materials have been revised.

“When utilized in osmotic energy conversion, a superior power density of 22.2 W m⁻², and a record-high selectivity of 0.975 and thus an exceptional energy conversion efficiency of 45.12% are simultaneously achieved, outperforming the majority of existing materials.” has been revised to “When utilized in osmotic energy conversion, a superior power density of 21.2 W m⁻², and a record-high selectivity of 0.976 and thus an exceptional energy conversion efficiency of 45.3% are simultaneously achieved, outperforming the majority of existing materials.”

“By mixing artificial sea water and river water, the 3D SCOF membranes exhibited a superior power density of 22.2 W m⁻², more than 4-fold larger than the commercialization benchmark, and a record-high ion selectivity of 0.975 and thus an exceptional energy conversion efficiency of 45.12% simultaneously. Furthermore, the power density reached 71.8 W m⁻² under a 500-fold salinity concentration gradient,

indicating the membrane applicability in hypersaline environments.” has been revised to “By mixing artificial sea water and river water, the 3D SCOF membranes exhibited a superior power density of 21.2 W m^{-2} , more than 4-fold larger than the commercialization benchmark, and a record-high ion selectivity of 0.976 and thus an exceptional energy conversion efficiency of 45.3% simultaneously. Furthermore, the power density reached 69.6 W m^{-2} under a 500-fold salinity concentration gradient, indicating the membrane applicability in hypersaline environments.”

“The net ionic current of $24.4 \mu\text{A}$ at $V = 0$ indicated a net cation flux from the high-concentration side to the low-concentration side and reflected a cation selectivity of the 3D SCOF membrane. The values of the diffusion potential (E_{diff}) can be obtained after subtracting the redox potential from the measured open-circuit voltage. For the 3D SCOF membrane, the E_{diff} value reached 95.473 mV , which was much higher than those of previously reported membranes. Furthermore, the corresponding cation selectivity (quantitatively described by cation transference number, t) was calculated to be 0.975, which was so close to the ideal cation selectivity ($t = 1$).” has been revised to “The net ionic current of $4.8 \mu\text{A}$ at $V = 0$ indicated a net cation flux from the high-concentration side to the low-concentration side and reflected a cation selectivity of the 3D SCOF membrane. A salt bridges was used to eliminate the redox potential of the electrodes, so we could directly obtain the diffusion potential (E_{diff}) from the intercept on the voltage axe of the I–V curves. The E_{diff} value of the 3D SCOF membrane reached 88.9 mV , which was much higher than those of previously reported membranes. Furthermore, the corresponding cation selectivity (quantitatively described by cation transference

number, t) was calculated to be 0.976, which was close to the ideal cation selectivity ($t = 1$).”

“Therefore, the corresponding energy conversion efficiency achieved up to 45.12%, which is the highest values reported.” has been revised to “Therefore, the corresponding energy conversion efficiency achieved up to 45.3%, which is the highest value reported.”

“The power density reached a maximum value of 22.2 W m^{-2} at a load resistance of $7.4 \text{ k}\Omega$ (comparable to the internal resistance of the device), which exceeded most reported 1D and 2D nanofluidic channel membrane systems to date and was over four times the value of the commercialization benchmark of $\sim 5 \text{ W m}^{-2}$.” has been revised to “The power density reached a maximum value of 21.2 W m^{-2} at a load resistance of about $7 \text{ k}\Omega$ (comparable to the internal resistance of the device), which exceeded most reported 1D and 2D nanofluidic channel membrane systems to date and was over four times the value of the commercialization benchmark of $\sim 5 \text{ W m}^{-2}$.”

“Nafion membrane exhibited a peak power density of 7.2 W m^{-2} , much lower than that of 3D SCOF membrane, confirming the crucial role of abundant interconnected channels.” has been revised to “Nafion membrane exhibited a peak power density of 6.4 W m^{-2} , much lower than that of 3D SCOF membrane, confirming the crucial role of abundant interconnected channels.”

“The maximum power density can achieve an amazing value of $\sim 71.8 \text{ W m}^{-2}$ at the 500-fold concentration gradient, indicating the application viability of the 3D SCOF membrane in hypersaline environments.” has been revised to “The maximum power density can achieve $\sim 69.6 \text{ W m}^{-2}$ at the 500-fold concentration gradient, indicating the

application viability of the 3D SCOF membrane in hypersaline environments.”

“It was found that the power densities follow the general trend of KCl (28.6 W m^{-2}) > NaCl (22.2 W m^{-2}) > LiCl (15.5 W m^{-2}) > CaCl₂ (13.8 W m^{-2}), which was consistent with the order of diffusion coefficients of cations (Supplementary Fig. 26).” has been revised to “It was found that the power densities follow the trend of KCl (28.6 W m^{-2}) > NaCl (21.2 W m^{-2}) > LiCl (15.5 W m^{-2}) > CaCl₂ (13.8 W m^{-2}), which was consistent with the order of diffusion coefficients of cations (Supplementary Fig. 26).”

“Our 3D SCOF membrane showed a superior energy conversion efficiency of 45.12%, which was the highest among all existing materials utilized in osmotic energy conversion, with a simultaneously high output power density of 22.2 W m^{-2} , outperforming most of state-of-the-art membranes.” has been revised to “Our 3D SCOF membrane showed a superior energy conversion efficiency of 45.3%, which was the highest among all existing materials utilized in osmotic energy conversion, with a simultaneously high output power density of 21.2 W m^{-2} , outperforming most of state-of-the-art membranes.”

“When utilized in osmotic energy conversion, the maximum output power density was 22.2 W m^{-2} , the ion selectivity was 0.975 and the energy conversion efficiency was 45.12% under a 50-fold salinity gradient between seawater and river water, outperforming most state-of-the-art membranes.” has been revised to “When utilized in osmotic energy conversion, the maximum output power density was 21.2 W m^{-2} , the ion selectivity was 0.976 and the energy conversion efficiency was 45.3% under a 50-fold salinity gradient between seawater and river water, outperforming most state-of-

the-art membranes.”

The original Fig. 3 has been revised to

Fig. 3 Proton conduction and osmotic energy conversion performance of 3D SCOF

membrane. a Temperature-dependent proton conductivity of the 3D SCOF membrane

under 100% RH. **b** Comparison in proton conductivity of the COFs or MOFs-based

proton conductors, polymer-based PEMs in literature and 3D SCOF membrane in our

work. Corresponding ion-conducting materials were shown in Supplementary Table 3.

c Schematic of the osmotic energy conversion process. **d** I–V curve of the 3D SCOF

membrane under a 0.01 M/0.5 M NaCl concentration gradient. **e** Current density and power density of 3D SCOF membrane as functions of external resistances under a 50-fold concentration gradient. **f** Output power density and energy conversion efficiency of the 3D SCOF membrane compared with the state-of-the-art membranes. Corresponding membranes were shown in Supplementary Table 4.

In the “**Energy conversion efficiency**” section of the Supplementary Materials, “The diffusion potential (E_{diff}) can be calculated as:

$$E_{diff} = V_{oc} - E_{redox}$$

$$E_{redox} = \frac{RT}{zF} \ln \frac{\gamma_{CH} C_H}{\gamma_{CL} C_L}$$

where V_{oc} , E_{redox} , R , T , F , z , γ and c refer to the open-circuit voltage, redox potential, universal gas constant, absolute temperature, Faraday constant, charge number, activity coefficient of ions, and ion concentration, respectively.

For a given concentration gradient, the cation transference number (t) can be calculated as:

$$t = \frac{1}{2} \left(\frac{E_{diff}}{\frac{RT}{zF} \ln \left(\frac{\gamma_{CH} C_H}{\gamma_{CL} C_L} \right)} + 1 \right)$$

” has been revised to

“For diffusion potential (E_{diff}) of 3D COF membranes in asymmetric electrolytes (0.01M/0.5M NaCl solution), a salt bridge was used to eliminate the redox potential from Ag/AgCl electrodes (commercial Ag/AgCl wire electrodes, diameter: ~ 0.5 mm, charged with salt bridges).

E_{diff} can be calculated as: $E_{diff} = V_{oc} - E_{redox}$, where V_{oc} and E_{redox} refer to the open-circuit voltage and redox potential, respectively. The generation of E_{redox} is

avoided by connecting the electrodes to the cell with agarose salt bridges. Therefore, the E_{diff} value is directly obtained from the intercept on the voltage axis of the current-voltage (I-V) curves.

For a given concentration gradient, the cation transference number (t) can be calculated as:

$$t = \frac{1}{2} \left(\frac{RT}{zF} \frac{E_{diff}}{\ln\left(\frac{\gamma_{CH} C_H}{\gamma_{CL} C_L}\right)} + 1 \right)$$

where R , T , F , z , γ and c refer to the universal gas constant, absolute temperature, Faraday constant, charge number, activity coefficient of ions, and ion concentration, respectively.”

The original Supplementary Fig. 25 has been revised to

Supplementary Fig. 25 (a) The current density and power density of 3D SCOF

membrane as functions of external resistances under a 5-fold concentration gradient; (b) Current density and power density of 3D SCOF and Nafion membranes as functions of external resistances under a 50-fold concentration gradient; (c) Current density and power density of 3D SCOF membrane as functions of external resistances under a 500-fold concentration gradient; (d) Output power density of the 3D SCOF membranes under different concentration gradient.

The original Supplementary Fig. 26 has been revised to

Supplementary Fig. 26 The output power density of the 3D SCOF membranes under other typical types of chloride salt solutions.

In the original Supplementary Table 4, the osmotic energy conversion performance of our 3D SCOF membrane has been revised.

Reviewers' Comments:

Reviewer #3:

Remarks to the Author:

Thanks for your serious responses.

"Moreover, to minimize any influence from unexpected factors, we implemented an agarose salt bridge to connect the electrodes to the cell, which can eliminate the redox potential of the electrodes." The authors should give the photograph of the experimental device to replace the Supplementary Fig. 23, before the publication of this manuscript. This article requires no further review.

Response to reviewers

Reviewer (Remarks to the Author):

Thanks for your serious responses.

"Moreover, to minimize any influence from unexpected factors, we implemented an agarose salt bridge to connect the electrodes to the cell, which can eliminate the redox potential of the electrodes." The authors should give the photograph of the experimental device to replace the Supplementary Fig. 23, before the publication of this manuscript.

This article requires no further review.

Reply: Thank the reviewer for the helpful guidance on our manuscript.

Based on the reviewer's valuable guidance, the photograph of the experimental setup device for current-voltage ($I-V$) measurement has been provided in the Supplementary Information as Supplementary Fig. 23. The original Supplementary Fig. 23, the photograph of the experimental setup device for osmotic energy conversion performance test, also remains in the Supplementary Information as Supplementary Fig. 24.

Supplementary Fig. 23 Photograph of the experimental setup device for current-

voltage (I - V) measurement. The device is connected to the external detection equipment for collecting output current and voltage.

The corresponding description in the main text has been revised.